# Full Length Transcriptome Highlights the Coordination of Plastid Transcript Processing

**DOI:** 10.3390/ijms222011297

**Published:** 2021-10-19

**Authors:** Marine Guilcher, Arnaud Liehrmann, Chloé Seyman, Thomas Blein, Guillem Rigaill, Benoit Castandet, Etienne Delannoy

**Affiliations:** 1Institute of Plant Sciences Paris-Saclay (IPS2), Université Paris-Saclay, CNRS, INRAE, Université Evry, 91405 Orsay, France; marine.guilcher@universite-paris-saclay.fr (M.G.); arnaud.lieh@gmail.com (A.L.); thomas.blein@cnrs.fr (T.B.); guillem.rigaill@inrae.fr (G.R.); benoit.castandet@universite-paris-saclay.fr (B.C.); 2Institute of Plant Sciences Paris-Saclay (IPS2), Université de Paris, CNRS, INRAE, 91405 Orsay, France; 3Laboratoire de Mathématiques et de Modélisation d’Evry (LaMME), Université d’Evry-Val-d’Essonne, UMR CNRS 8071, ENSIIE, USC INRAE, 91000 Evry, France; chloeseyman@gmail.com

**Keywords:** *Arabidopsis thaliana*, plastid, co-maturation, post-transcriptional, nanopore

## Abstract

Plastid gene expression involves many post-transcriptional maturation steps resulting in a complex transcriptome composed of multiple isoforms. Although short-read RNA-Seq has considerably improved our understanding of the molecular mechanisms controlling these processes, it is unable to sequence full-length transcripts. This information is crucial, however, when it comes to understanding the interplay between the various steps of plastid gene expression. Here, we describe a protocol to study the plastid transcriptome using nanopore sequencing. In the leaf of *Arabidopsis thaliana*, with about 1.5 million strand-specific reads mapped to the chloroplast genome, we could recapitulate most of the complexity of the plastid transcriptome (polygenic transcripts, multiple isoforms associated with post-transcriptional processing) using virtual Northern blots. Even if the transcripts longer than about 2500 nucleotides were missing, the study of the co-occurrence of editing and splicing events identified 42 pairs of events that were not occurring independently. This study also highlighted a preferential chronology of maturation events with splicing happening after most sites were edited.

## 1. Introduction

Plastids are derived from the endosymbiosis between photosynthetic organisms and an ancestral Eukaryote. Although most of the initial symbiont genes have been transferred to the nucleus during the course of evolution, plastids of land plants and other photosynthetic Eukaryotes still maintain a small but essential genome. It mainly encodes subunits of each of the photosynthetic complexes (Photosystem I and II, cytochrome b6/f, ATP synthase and Rubisco) and some of the plastid gene expression (PGE) machinery [1]. Most of the proteins involved in PGE are, however, encoded in the nucleus and need to be targeted back to plastids. As a consequence, PGE retains characteristics from both eukaryotes and bacterial systems, resulting in a sophisticated interplay between nucleus and plastid encoded factors [2,3,4].

A striking feature of PGE is the importance and complexity of the post-transcriptional maturation steps. In addition to the intron removal by RNA splicing [5] and the specific conversion of cytosines into uridines by RNA editing [6], complete maturation also requires intergenic cleavage of the multigenic transcripts and the generation of 5′ and 3′ ends through RNA processing [7,8]. Most of the RNA binding proteins (RBP) or ribonucleases known to be involved in PGE are localized in a membraneless structure surrounding the plastome—the nucleoid [9]. This close association between RNA maturation factors might be an explanation for the multiple pleiotropic effects observed in chloroplast mutants [7].

Various investigations, both in vitro and in organellar gene expression mutant plants, have indeed revealed situations where the different maturation events can influence each other. For example, intron removal is a prerequisite for editing in the *ndhA* second exon [10] and *atpF* splicing is severely reduced in the *aef1* mutant in which the editing of *atpF_12707* is abolished [11]. *Arabidopsis thaliana* chloroplast RNA editing is affected in a mutant deficient for the exoribonuclease PNPase [12] while correct processing of the potato mitochondrial tRNA Phe requires RNA editing [13]. Editing sites can even influence each other. For example, in *A. thaliana*, editing of mitochondrial *ccmB_17869* by MEF19 depends on the editing of *ccmB_17884* by MEF37 [14]. Similarly, in *Physcomitrium patens,* editing of the mitochondrial *ccmFc-C103* by PpPPR_65 controls editing of *ccmFc-C122* by PpPPR_71 [15,16].

These dependencies are usually explained according to two models. First, one maturation event can modify the RNA secondary structure necessary for the second maturation. Second, the proteins responsible for the maturation can interact with each other or, more directly, target several maturation events. Most studies, however, only focused on a limited set of transcripts or RNA maturation events precluding any general conclusions. This illustrates the urgent need for the development of global approaches capable of simultaneously studying all the RNA maturation processes, at the transcriptomic level. This issue has recently been tackled by the increasing use of Illumina-based RNA-Seq strategies to study PGE from transcription to translation [17,18,19,20,21,22,23].

Although this has considerably increased the power and sensitivity of PGE analyses, it is ill-suited to study the potential coordination between maturation steps. The short reads used by Illumina technology (the maximum insert size of Illumina TruSeq RNA libraries reaches around 350 base-pairs) make it impossible to monitor the co-occurrence of these events on single RNA transcripts that can be several kilobases long. An alternative would be to take advantage of other sequencing technologies such as PacBio or Oxford Nanopore. They theoretically allow the sequencing of full-length cDNAs or RNA and should therefore overcome the current technical limitations [24]. A major issue, however, is that most of the available library preparation protocols only capture polyadenylated RNA transcripts, therefore excluding plastid transcripts. A recent protocol analyzing chromatin-bound transcripts also captures non-polyadenylated transcripts but was not applied to the analysis of plastid transcripts [25,26].

In this work, we describe the analysis of the *A. thaliana* plastid transcriptome by sequencing full-length non-polyadenylated and polyadenylated cDNAs using the Oxford Nanopore technology (ONT). This analysis identified all known post-transcriptional maturation events and provided an overview of their coordination in normal growth conditions.

## 2. Results

### 2.1. A Protocol to Sequence the Full Length Plastid Transcriptome

The library synthesis protocol is derived from the Switching Mechanism at the 5′ end of RNA Transcript (SMART) technology developed to synthesize full-length cDNAs [27]. Because polyadenylation of chloroplastic RNAs acts as a degradation signal [28], we, however, had to first start with the ligation of an RNA adapter (modified from Hotto et al. [29]) at the 3′ end of the RNAs to allow the priming of the reverse transcription and an rRNA depletion before completing the cDNA synthesis. The cDNAs are then incorporated into an ONT sequencing library and sequenced. Sampling RNA from leaves of 5 week-old col-0 *A. thaliana* plants grown in long-day conditions at 20 °C, we mapped between 1.55 million and 2.69 million stranded reads (mapping rate between 98.5% and 99.8%) to the *A. thaliana* genome including between 10% and 40% to the plastid genome and between 0.3% and 0.8% to the mitochondrial genome. The median error rate was between 4% and 4.4%. The rRNA depletion was very efficient with less than 0.1% of reads mapping to rRNA loci. More than 99.5% of the reads mapped to the annotated nuclear genes corresponding to the sense orientation, a proportion similar to Illumina stranded RNA-Seq. Most of the reads (99%) were between 195 and 2141 nucleotides (nt) long with a median size of 852 nt and a maximum size of 4805 nt. In *A. thaliana*, 7261 genes are producing transcripts longer than 2141 nt and more than 390 genes (including the plastid *ycf2* gene) are producing transcripts longer than 4800 nt. Based on the whole transcriptome, the 3′ to 5′ transcript coverage was better with our protocol than for similar samples analyzed using Illumina sequencing for transcripts below 1500 nt (22,853 genes; Appendix A). For transcripts above 1500 nt (17,985 genes), the Illumina sequencing performed better and a moderate 3′-5′ bias can be observed. These results confirm that our nanopore reads were mostly full-length and stranded but that the longer transcripts are missing from the sequencing libraries.

### 2.2. A Representative Picture of the Plastid Transcriptome

With at least 275,000 reads mapped on the plastid genome for each biological replicate, the coverage is deep enough to have a good representation of the plastid transcriptome. To verify that the sequencing data are correctly capturing the plastid transcriptome, we looked at the complex transcriptional profile of the *psbB* to *petD* genomic region (Figure 1).

Following transcription, transcripts from this multigenic locus are processed into multiple poly- or monocistronic isoforms on both genomic strands [30,31]. A rapid overview of the reads showed the transcription of *psbN* on the Crick strand while *psbB*, *psbT*, *psbH*, *petB* and *petD* were transcribed from the Watson strand as expected. The spliced *petD* and *petB* transcripts were also found. Taking advantage of long-read sequencing, it is possible to emulate Northern blots by selecting reads which map on specific positions and plotting the distribution of the read lengths. Felder et al. [30] studied the involvement of HCF107 in the processing of the *psbB* to *petD* locus with an extensive use of Northern blots, allowing a comparison of the two methods. We therefore generated virtual Northern blots for *psbN*, *psbH*, *petB* and *petD* (Figure 2) using virtual probes equivalent to the probes used for Figure 4C,E,H,I of Felder et al. [30].

Reads mapping to *psbN* were almost exclusively 200 nt long which is compatible with the signal detected by a classic Northern blot. Reads mapping to *psbH* showed two major isoforms around 1100 nucleotides (nt) and 1800 nt but also two minor isoforms around 370 nt and 2600 nt. This profile is also compatible with the regular Northern blot. However, Felder et al. [30] also detected larger isoforms at 3300, 4100, 4900, and 5600 nt that were not captured in our sequencing libraries. The virtual Northern blot for *petB* showed four major isoforms at 750, 1100, 1450, and 1800 nt. A faint isoform may be present at 2250 nt. These isoforms were also detected by Felder et al. [30] who found additional isoforms at 2600, 3300, 4100, 4900, and 5600 nt. Finally, for *petD*, we found two major isoforms around 1450 and 1800 nt and minor isoforms around 990 and 2225 nt. We missed the larger isoforms detected by Felder et al. [30] but also a 1200 nt isoform described as an unspliced *petD* transcript which seemed to be replaced by our 990 nt isoform. The detection of sharp “bands” of the expected size in our virtual Northern blots confirms that the majority of the nanopore reads correspond to full-length cDNAs but this result also confirms that transcripts longer than 2–2.5 kb are under-represented in our sequencing libraries.

In these complex loci, it is sometimes difficult to identify all the bands on a regular Northern blot. For example, Felder et al. did not associate the 2200 nt transcript of their *petB* and *petD* Northern blots to a particular isoform.

Our sequencing showed that this transcript is most likely a polycistronic intermediate containing an unspliced *petB* with a spliced *petD* (Figure 3A). For *petD*, we detected a minor isoform around 990 nt. The associated transcripts corresponded to two distinct isoforms (Figure 3B). The first one corresponded to spliced *petD* transcripts but with 5′ ends within the second *petB* exon. The second one had a 5′ end in the *petD* intron at position 76,780 and included the second *petD* exon. Position 76,780 was identified as a transcription start site and multiple 5′ ends were mapped in this area [20]. Similarly, because of their poor resolution, regular Northern blots can miss isoforms of similar sizes. Our virtual Northern blot for *psbH* showed that the four peaks are double peaks: the main isoforms are each associated with isoforms which are 50 nt longer. When mapping these isoforms, we could show that the short and long isoforms are associated with different 5′ ends, the long one around the genomic position 74,393 and the short one around 74,441 (Figure 3C). According to Castandet et al. [20], position 74,441 corresponds to the major processed extremity of *psbH* while position 74,393 is a transcription start site.

Even if our nanopore reads showed no or only moderate 3′ to 5′ bias in general (Appendix A), some plastid transcripts showed 3′ to 5′ but also 5′ to 3′ biases (Figure 4).

**Figure 3 ijms-22-11297-f003:**
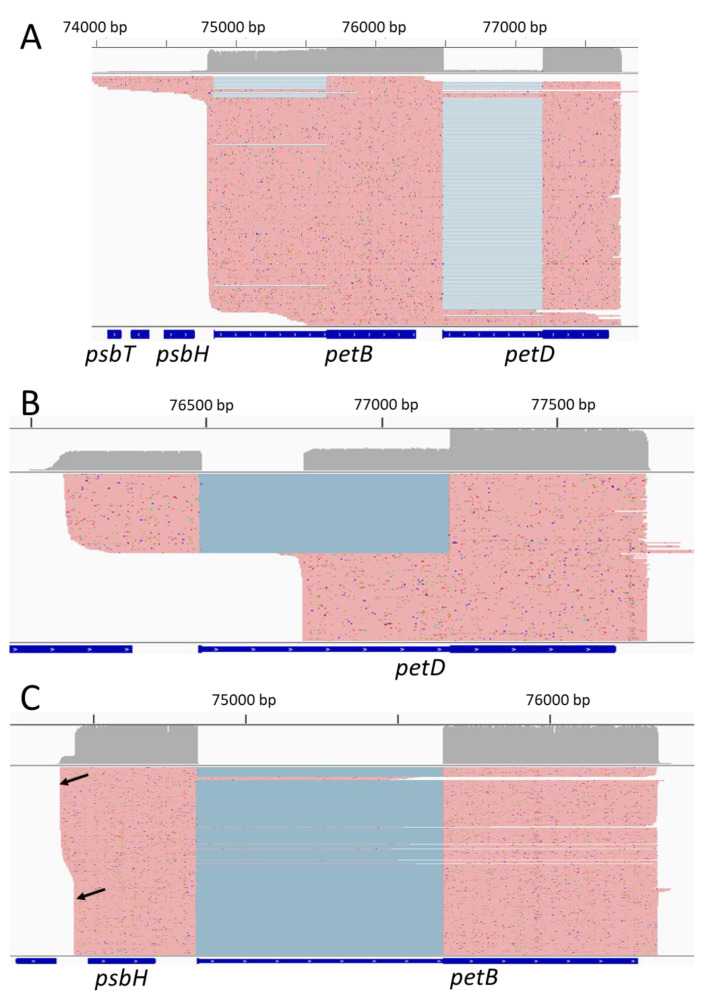
Identification of transcripts isoforms. Screenshots of IGV displaying the reads corresponding to various virtual Northern blot isoforms. Matching bases are shown in red. Split reads are joined by blue lines. Other colors indicate mismatches and indels. (**A**) Reads corresponding to the 2200 nt isoform of the *petB* and *petD* virtual Northern blots. (**B**) Reads corresponding to the 990 nt isoform of the *petD* virtual Northern blot. (**C**) Reads corresponding to the 1100–1150 isoform of the *psbH* virtual Northern blot. The two 5′ ends are shown by black arrows.

**Figure 4 ijms-22-11297-f004:**
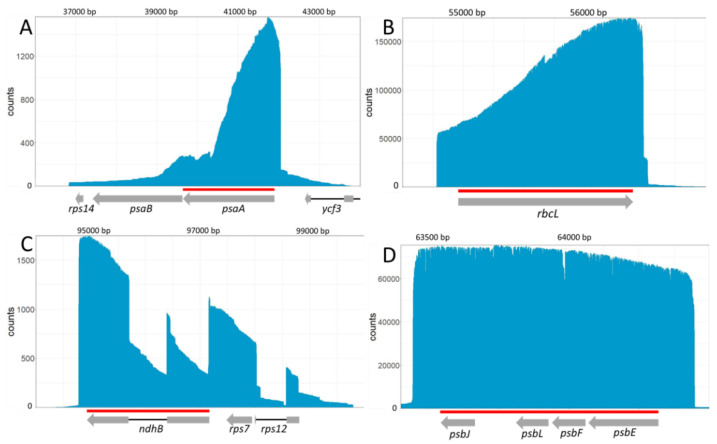
Examples of coverage biases in plastid transcripts. For each panel, the coverage at single-nucleotide resolution by strand-specific reads overlapping at least partially the genomic regions shown in red is shown. At the top, the genomic positions are shown while the coding sequences associated with these regions are shown below as gray arrows or boxes. Introns are represented as black lines. (**A**): *psaA* transcripts. (**B**): *rbcL* transcripts. (**C**): *ndhB* transcripts. (**D**): *psbE-psbJ* transcripts.

In *ndhB*, the coverage at the 3′ end was only about 20% of the coverage at the 5′ end. This bias could be technical but it was not the same for other transcripts (for example *rbcL* or the polycistronic *psbE-F-L-J* transcript). For *psaA*, we observed a strong 5′ to 3′ bias. It may illustrate the pattern of transcript degradation of the 5300 nt long *psaA-psaB-rps14* transcript [32] which is absent of our sequencing libraries.

Finally, post-transcriptional maturation events can be quantitatively analyzed. Known editing sites could be detected with rates comparable to leaf datasets (Table 1, Appendix A) previously published by Guillaumot et al. [33] (Pearson correlation = 0.97; *p*-value < 2.2 × 10^−16^) and Ruwe et al. [12] (Pearson correlation = 0.94; *p*-value < 2.2 × 10^−16^).

It should be noted that the analysis of poorly edited sites by nanopore sequencing must be done carefully because of the relatively high error rate of this technology. For example, using the same pipeline as Guillaumot et al., we detected 123 plastid C to U transitions with a rate higher than 10% but only 44 of them were also detected by Guillaumot et al. [33] using Illumina sequencing. Similarly, intron splicing efficiency could be measured (Table 1 and Appendix A) and it varied from 4% to 97% depending on the intron. Most values are higher (by 22 points on average) than the efficiencies measured by Guillaumot et al. [33] using Illumina. This bias could be explained by an under-representation of long (unspliced) transcripts compared to short (spliced) ones. However, this bias is not linked to the unspliced transcript length, the intron length or the unspliced/spliced size ratio (Appendix A). An alternative, but not exclusive, explanation is that the abundance of unspliced transcripts is difficult to estimate with Illumina sequencing.

### 2.3. Some Post-Transcriptional Events Are Coordinated and Ordered

Because editing and splicing events are well defined (a single genomic position, either processed or not), it is easy to statistically analyze the possible coordination between these events.

Although 1596 co-occurring events could theoretically be expected with 14 splicing and 43 editing events analyzed, only 138 co-occurrences were detected at least once. This is, however, expected as all events are not found on a single transcript (Appendix A). Out of these 138 pairs of maturation events, 42 were not found to occur independently (Figure 5). Conversely, we did not detect any complete dependency (when one maturation event is absolutely required for another maturation event to occur). We observed partial dependencies between splicing events (*clpP* introns, *petD* and *petB* introns), editing and splicing events (in the *atpF*, *clpP* and *ndhB* transcripts) and between editing events (in the *rps14, ndhD* and *ndhB* transcripts). This partial dependency also occurred between different genes (*petD* and *petB*; *psbE* and *psbF*) belonging to polycistronic transcripts. Some sites of coordinated events like *ndhD_116290* and *ndhD_116281*, *ndhB_95650* and *ndhB_95644 or ndhB_95419* and the *ndhB* intron could be very close but the others were separated by more than 100 nt.

A more detailed analysis shows that maturation intermediates (TF (True-False) and FT (False-True) columns of Appendix A) were always less frequent than expected for independent events for the 42 pairs of dependent events. This means that when one site was processed the second one was more processed than expected randomly. In other words, there was co-maturation but no incompatibility, the maturation of one site increased the rate/speed of maturation of the other one. Furthermore, comparing the abundance of the intermediates of maturation offers the opportunity to find a preferred order of the maturation events (Figure 6).

This analysis suggests that RNA editing at *psbE_64109* generally occurred before RNA editing at *psbF_63985*, and that the splicing of *petD* preferentially occurred before the splicing of *petB*. The maturation of *clpP* generally started with RNA editing at *clpP_69942* followed by the splicing of the second intron and finished by the splicing of the first intron. For *ndhD*, the maturation preferentially started with RNA editing at *ndhD_116785* followed by *ndhD_116494* then both *ndhD_116290* and *ndhD_116281* to finish with *ndhD_117166*, the editing site creating the start codon of *ndhD*. For the *ndhB* transcript, the chronology of the maturation seemed more convoluted as three sites (96,457, 96,439 and 95,225) were edited independently of the other maturation events. RNA editing at *ndhB_97016* seemed to occur first followed by editing at the four sites 96,579, 95,650, 95,608 and 94,999. The maturation of *ndhB* ended with RNA editing at sites 96,698, 96,419, 95,644 and, probably slightly later, its splicing. To confirm the order deduced from the co-occurrence analysis for transcripts requiring more than three maturation events (i.e., *ndhD* and *ndhB*), we identified the reads covering all the maturation events and counted the frequency of the various intermediates (Appendix A). Out of 413 intermediate reads, 311 (75%) were compatible with the proposed chronology of *ndhD* maturation. For *ndhB*, only 63 intermediate reads were identified. This number is too small to estimate the frequency of the 4096 possible intermediates (12 maturation events) but 35 (57%) were compatible with the proposed chronology. The reads corresponding to alternative maturation chronologies are probably the result of sequencing errors. Given the nanopore sequencing error rate at around 4% and the fact that this analysis considered five positions in *ndhD* and 12 in *ndhB*, only 81.6% (0.96^5^) of the *ndhD* reads and 62% (0.96^12^) of the *ndhB* reads are expected to be error-free at these positions.

This preferred chronology could theoretically be the result of kinetic differences between the different maturation events. For example, looking at two independent events, the one happening at a higher rate will likely occur first. This simple explanation is, however, incompatible with the observations. In particular, the decrease of the observed vs. expected TF and FT counts (columns delta_TF and delta_FT in Appendix A) is not homogenous between TF and FT for most pairs of events. This shows that the positive effect (e.g., enhancement) provided by one maturation event to the other is not symmetrical. This asymmetry is involved in the chronology and could reinforce (at least in this case) any putative effects caused by a difference in processing rates. Finally, because of the number of maturation events jointly monitored for the *ndhB* (12 events) and *ndhD* transcripts (5 events, Appendix A), the observation of a preferred chronology of maturation is extremely unlikely to be explained only by differences in maturation speed. We conclude that the observed preferred chronology of maturation is due, at least partly, to interactions between the processing events.

## 3. Discussion

Our protocol generates mostly full-length and stranded reads but transcripts longer than 2000–2500 nt are clearly under-represented. This bias is common to nuclear and plastid transcripts and several pieces of evidence (data not shown) strongly suggest that it is associated with the initial RNA-RNA ligation at the 3′ end of transcripts. It has indeed been described that the ligation step was sensitive to secondary structures at the 3′ end [35]. Maybe the denaturation step preceding the ligation step was not sufficient for long transcripts.

Following transcription, plastid transcripts undergo a complex array of modifications and maturation and the recent massive use of RNA-Seq based strategies has led to an unprecedented knowledge about its different steps. What is sorely lacking, however, is a global understanding of the interplay between RNA editing, splicing and processing.

Initially thought to be mainly independent [36,37], there are now more and more pieces of evidence for crosstalk between the different maturation steps [10,38,39,40,41]. Most of these results, however, have been obtained from experiments based on Sanger sequencing of a cDNA of interest, therefore limiting any potential generalization. Taking advantage of the development of nanopore sequencing, we systematically studied the link between individual RNA splicing and RNA editing events, at the plastome level.

Our results show that co-maturation of several sites tends to occur even when located far apart on their cognate transcript. This implies that all of the actors of these different processing events are grouped or co-localized, likely in the nucleoid [9].

Looking at specific links, splicing of the *atpF* intron and RNA editing at the *atpF_12707* site are clearly not independent (Figure 5). This was expected as AEF1, the PPR protein responsible for *atpF_12707* editing in *A. thaliana*, also facilitates *atpF* splicing [11]. Similarly, *clpP* intron 2 and *ndhB* splicing is enhanced by RNA editing in the cognate transcripts (Figure 6). Earlier studies have shown that some unspliced or unprocessed transcripts can already be fully edited [36,37] and this was interpreted as evidence that RNA editing is an early process, mainly occurring before splicing. Although RNA editing can be a prerequisite for splicing when it restores sequences or structures within the intron [42,43], this is an unlikely explanation here as the sites are located far from the identified splicing key elements [44]. A possibility put forward by Yap et al. [11] is that the binding of the RNA editing factor itself could have an indirect effect on splicing through the modification of RNA secondary structure or accessibility.

In agreement with the idea that RNA editing is an early maturation step, we only found marginal evidence that specific RNA editing sites could be influenced by splicing (Figure 5). This result is, however, probably dependent on our experimental model, *A. thaliana*. In various plants, *ndhA* intron removal was shown to be necessary for a *ndhA* editing site located close to the 3′ splice site. In this case, splicing is thought to create the RNA sequence necessary for the recognition of the RNA editing site [10], a site that is absent in *A. thaliana*. A similar situation has been described in *P. patens* mitochondria where *atp9* splicing is necessary to one editing site on the same transcript [15]. As shown for *clpP*, the splicing of one intron can also influence the splicing of another intron located on the same transcript (Figure 5 and Figure 6). Experiments with intron deletions in tobacco have previously shown that the second intron in the *ycf3* transcript needs to be spliced before the first intron. In this case, splicing of the first intron was hypothesized to create a sequence masking essential structural elements of the second intron [45]. Although *A. thaliana ycf3* structure is similar to tobacco, our analysis did not confirm such dependence in this transcript.

The dependence between RNA editing sites themselves has long been debated. For example, in vitro results on short fragments of the mitochondrial *atp4* RNA suggested that editing of individual sites did not influence others while *in organello* experiments with longer *cox2* transcripts showed a pattern of dependencies [46,47]. The identification of distal elements able to enhance RNA editing was also a strong argument against complete stochastic independence of the editing site recognition [41,48]. Our results show that both cases exist in the chloroplast. Editing site *ndhD_117166* generally requires earlier editing of the four other *ndhD* sites and *ndhB_97016* editing strongly influences editing at *ndhB_96698* and *ndhB_96579* sites. On the other hand editing at *ndhB_95225* seems autonomous and barely influences any other editing site (Figure 6).

Editing and splicing of organellar transcripts are required to get mRNA translated into functional proteins as editing often restores conserved amino acids [49] and splicing preserves the translation frame. However, the study of the translational landscape of *A. thaliana* mitochondria [50] or maize chloroplasts [21] showed that ribosomes were associated with partially edited transcripts and a small fraction of ribosomes were even associated with intronic sequences. Earlier chloroplast polysome purification experiments also showed that transcripts of the *psbB* gene cluster containing the *petB* or *petD* intron could still be translated for other genes [51]. This suggests that partially mature (especially partially edited) transcripts can access the organelle translation machinery. In addition to the dependence of some maturations events, our results showed that they could be ordered (Figure 6). In this chronology, splicing events seemed to occur later than editing events: the splicing of *ndhB* occurred after editing at most sites and splicing of *clpP* occurred after its editing. Even if the chronology was not clear from our results, Yap et al. also showed that *atpF* editing probably occurs before its splicing [11]. In addition, events located at the 5′ end of the transcripts tended to be later than the others. That is clearly the case for *clpP* and *ndhD* associated transcripts. In *ndhD*, RNA editing at *ndhD_117166* was generally the last maturation event and is required to create the start codon and thus to allow the translation of the transcript. This succession of the maturation events where splicing and 5′ end events tend to be last could be a way to ensure the complete (or at least a better) maturation of the transcripts before initiating their translation. Although there is currently no known underlying mechanism to support this hypothesis, it could at least explain why partially edited RNA editing sites are generally more edited in ribosome-associated RNAs than on the steady-state pool of transcripts [21,50]. In addition, it could also explain why sites restoring cryptic start codons have variable but often lower editing rates [49,52].

Despite the modest size of the dataset and its rather simple analysis, the results presented in this study highlight the potential of long-read RNA-Seq for the analysis of plastid and mitochondrial transcriptomes. Even if the molecular protocol still needs improvements to capture the longest transcripts, it provides access to the full complexity of this transcriptome and already showed numerous links between splicing and editing. For analytical reasons, we did not include the analysis of processing in this study but nanopore RNA-Seq is suited for this type of analysis (Figure 3) and we are developing the required bioinformatical and statistical tools. A potential improvement of our strategy would be to directly sequence the chloroplastic RNAs, without performing any cDNA synthesis. This would give access to the various epitranscriptomics marks [53] that are now known to be pervasive in chloroplastic RNAs [54] and whose interactions have, for example, been shown to be important in human diseases [55]. With this complete toolbox, we anticipate it will be possible to explore the impact of growth conditions and/or mutants or compare the nucleoid- or polysome-associated transcriptome to further decipher the molecular mechanisms controlling plastid but also mitochondrial gene expression.

## 4. Materials and Methods

### 4.1. Plant Growth and RNA Extraction

Col-0 plants were grown in soil in growth chambers with 16 h of light per day at 20 °C for 5 weeks. Fifteen minutes before the onset of lights, 2 adult leaves were flash-frozen in liquid nitrogen. Total RNA was extracted using Nucleozol (Macherey-Nagel, Hoerdt, France) followed by a purification with AMPure RNA XP beads (Beckman Coulter, Villepinte, France). Three independent experiments were performed to get three biological replicates.

### 4.2. Nanopore Sequencing

The step-by-step protocol for the construction of the sequencing library is available online at https://forgemia.inra.fr/guillem.rigaill/nanopore_chloro (accessed on 18 October 2021). Briefly, 10 fmoles of the RNA oligo /5Phos/rNrNrNrNrUrGrArArUrGrCrArArCrArCrUrUrCrUrGrUrArC/3InvdT/ (IDT Technologies, Leuven, Belgium) was ligated to the 3′ end of 100 ng of total RNA using 10 U of T4 RNA ligase 1 (NEB, Evry, France). Ligated RNA was depleted of rRNA using the QIAseq FastSelect -rRNA Plant Kit (QIAGEN, Les Ulis, France) before a full-length cDNA synthesis using the SMARTScribe™ Reverse Transcriptase (Takara, Saint Germain en Laye, France) and the oligos AAGCAGTGGTATCAACGCAGAGTACrGrG + G and AAGCAGTGGTATCAACGCAGAGTACGTACAGAAGTGTTGCATTC (IDT Technologies, Leuven, Belgium). Full-length cDNAs were amplified with the SeqAmp DNA Polymerase (Takara, Saint Germain en Laye, France) using the AAGCAGTGGTATCAACGCAGAGTAC primer and purified with AMPure XP beads (Beckman-Coulter, Villepinte, France). 35 fmoles of amplified cDNAs were converted to a nanopore sequencing library with the PCR barcoding kit (Oxford Nanopore Technologies, Oxford, UK) and then sequenced on an R10.3 MinIon flow-cell (Oxford Nanopore Technologies, Oxford, UK).

### 4.3. Bioinformatics and Statistical Analyses

The raw data were base-called and demultiplexed with Guppy v5.0.7 (Oxford Nanopore Technologies) using the dna_r10.3_450 bps_hac model. Reads were then oriented using the in-house script “fastq_processing.sh” which uses LAST v1179 [56] and CUTADAPT v2.10 [57] and is available online at https://forgemia.inra.fr/guillem.rigaill/nanopore_chloro (accessed on 18 October 2021). They were mapped on the col-0 genomic sequence with Minimap2 v2.1 [58]. Transcript body coverage and strandedness were measured with the RSeQC v3.0 package [59]. The Illumina samples used to compare were the dyw2_HE replicates 1 to 3 (NCBI GEO accession numbers GSM2677518, GSM2677519 and GSM2677520) from Guillaumot et al. [33]. The plants used for these samples were grown in the same growth chambers and the sequencing libraries were constructed with the Illumina TruSeq stranded total RNA with Ribozero plant kit.

The maturation events analyzed in this study are listed in Appendix A. They include the editing sites detected by Ruwe et al. [12] and the introns of protein-coding genes. The tRNA introns were omitted because the mature tRNAs are excluded from the sequencing library during sizing. This information is used to annotate each read for every maturation event according to three modalities: mature site, not mature site, and not read site. The latter allows taking insertions/deletions into account which are frequent in nanopore datasets. For each pair of events jointly observed the following configurations are listed and counted in a contingency table: mature/mature, mature/immature, immature/mature, and immature/immature. The dependency of two events, based on the contingency table, is tested using a Fisher exact test and the *p*-values were adjusted with an FDR [60]. Only pairs of events characterized by an adjusted *p*-value < 0.1 in at least 2 of the 3 replicates and an adjusted *p*-value < 0.005 on the pool of the 3 replicates were considered significant. Commented R scripts to annotate reads, create contingency table, perform Fisher’s exact tests and generate the result table are available online at https://forgemia.inra.fr/guillem.rigaill/nanopore_chloro (accessed on 18 October 2021).

The splicing and editing rates were measured from pooled reads of the 3 replicates. Virtual Northern blots were generated by extracting the length of the reads mapping from position 75700 to position 76000 on the Watson strand (*petB*), from position 77200 to position 77500 on the Watson strand (*petD*), from position 74487 to position 74706 on the Watson strand (*psbH*) or from position 74254 to position 74378 on the Crick strand (*psbN*) using samtools [61] and bedtools [62]. The size distributions were normalized by setting the value of the most abundant read length to 100. These distributions were converted into virtual Northern blots with the “vNB.py” python script available online at https://forgemia.inra.fr/guillem.rigaill/nanopore_chloro (accessed on 18 October 2021).

## Figures and Tables

**Figure 1 ijms-22-11297-f001:**
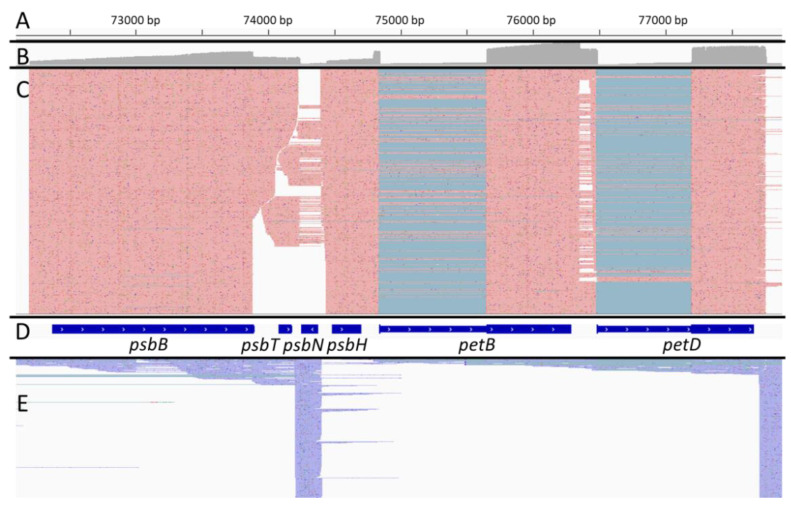
The complexity of the *psbB-petD* locus. Screenshots of Integrative Genomics Viewer (IGV) displaying nanopore reads mapping to the *psbB-petD* locus. (**A**) plastid genomic position. (**B**) coverage track displaying the number of reads at each nucleotide. (**C**) screenshot of reads mapping on the Watson strand. Matching bases are shown in red. Split reads are joined by blue lines. (**D**) Annotation of the locus. Introns are shown as thinner segments. (**E**) screenshot of reads mapping on the Crick strand. Matching bases are shown in purple. Split reads are joined by blue lines.

**Figure 2 ijms-22-11297-f002:**
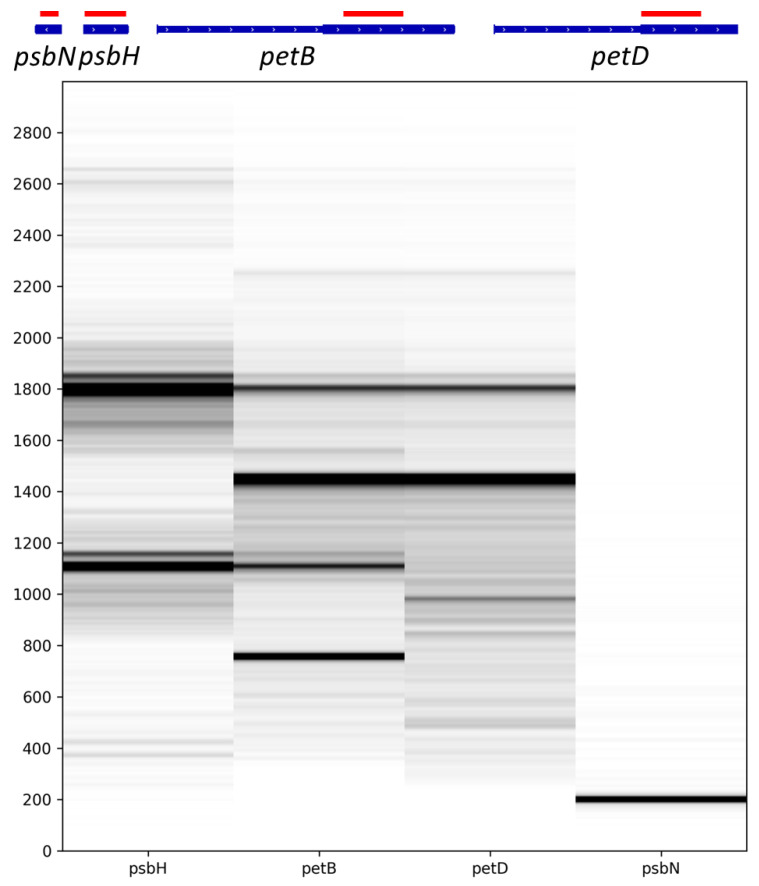
Virtual Northern blots derived from the nanopore sequencing. Northern blots were emulated from nanopore reads mapping to the sequences of *psbN*, *psbH*, the second exon of *petB*, or the second exon of *petD* shown in red on the genomic map displayed above. The size (in nt) is shown on the left.

**Figure 5 ijms-22-11297-f005:**
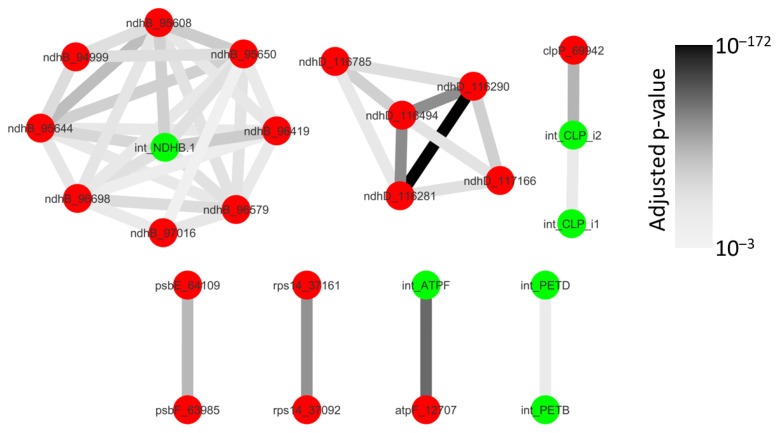
Network of splicing and editing coordination. Splicing events are shown in green and editing events in red. Dependent events are joined by an edge. The darkness of the edge is proportional to the adjusted *p*-value of the Exact Fisher test for the pool of the three replicates.

**Figure 6 ijms-22-11297-f006:**
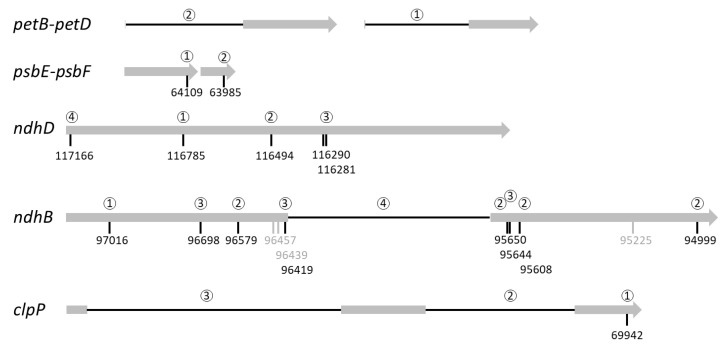
Proposed chronology of maturation events. Exons are shown as grey bars and introns as black lines. The editing sites are indicated by their genomic position. Grey editing sites are processed independently and thus are not included in the chronology. The preferred order of the maturation events is indicated by the numbers above the editing sites or introns.

**Table 1 ijms-22-11297-t001:** Quantification of known editing and splicing events.

Name	Type	Maturation Rate	Maturation Rate (Guillaumot et al., 2017)	Maturation Rate (Ruwe et al., 2013)
*int_RPS16*	splicing	4%	4%	NA
*int_ATPF*	splicing	89%	82%	NA
*int_RPOC1*	splicing	64%	19%	NA
*int_YCF3_i2*	splicing	79%	42%	NA
*int_YCF3_i1*	splicing	63%	45%	NA
*int_CLP_i2*	splicing	60%	71%	NA
*int_CLP_i1*	splicing	69%	62%	NA
*int_PETB*	splicing	91%	58%	NA
*int_PETD*	splicing	97%	62%	NA
*int_RPL16*	splicing	69%	12%	NA
*int_RPL2.1*	splicing	66%	52%	NA
*int_NDHB.1*	splicing	68%	55%	NA
*int_RPS12C*	splicing	92%	81%	NA
*int_NDHA*	splicing	68%	27%	NA
*matK_2931*	editing	53%	79%	93%
*atpF_12707*	editing	89%	91%	95%
*atpH_UTR_13210*	editing	5%	3%	4%
*rpoC1_21806*	editing	33%	21%	15%
*rpoB_23898*	editing	87%	82%	85%
*rpoB_25779*	editing	64%	83%	86%
*rpoB_25992*	editing	69%	76%	94%
*psbZ_35800*	editing	93%	90%	95%
*rps14_37092*	editing	89%	93%	94%
*rps14_37161*	editing	92%	97%	96%
*ycf3_i2_43350*	editing	16%	10%	12%
*rps4_UTR_45095*	editing	6%	3%	10%
*ndhK_ndhJ_49209*	editing	4%	4%	6%
*accD_57868*	editing	90%	95%	99%
*accD_58642*	editing	76%	75%	83%
*psbF_63985*	editing	90%	98%	98%
*psbE_64109*	editing	95%	100%	100%
*petL_65716*	editing	79%	91%	86%
*rps18_UTR_68453*	editing	3%	4%	NA
*rps12_69553*	editing	21%	26%	27%
*clpP_69942*	editing	82%	72%	81%
*rpoA_78691*	editing	78%	76%	91%
*rpl23_86055*	editing	34%	74%	75%
*ycf2_as_91535*	editing	3%	4%	NA
*ndhB_UTR_94622*	editing	8%	0%	NA
*ndhB_94999*	editing	88%	93%	94%
*ndhB_95225*	editing	95%	98%	99%
*ndhB_95608*	editing	87%	84%	80%
*ndhB_95644*	editing	78%	87%	81%
*ndhB_95650*	editing	88%	91%	84%
*ndhB_96419*	editing	75%	94%	92%
*ndhB_96439*	editing	6%	4%	6%
*ndhB_96457*	editing	6%	3%	5%
*ndhB_96579*	editing	90%	89%	90%
*ndhB_96698*	editing	81%	88%	82%
*ndhB_97016*	editing	94%	94%	95%
*ndhF_112349*	editing	85%	93%	96%
*ndhD_116281*	editing	76%	83%	92%
*ndhD_116290*	editing	77%	84%	90%
*ndhD_116494*	editing	88%	90%	93%
*ndhD_116785*	editing	94%	97%	98%
*ndhD_117166*	editing	35%	33%	45%
*ndhG_118858*	editing	69%	78%	85%

NA: Not Analyzed. The genomic position of each site and the corresponding nomenclature of Rüdinger et al. [34] are given in Appendix A.

## Data Availability

The fastq files are available from the NCBI SRA database under the accession number PRJNA748959.

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
