# Peer review of "Full Length Transcriptome Highlights the Coordination of Plastid Transcript Processing"

_ijms, 2021, doi:10.3390/ijms222011297_

Round 1
Reviewer 1 Report
In the manuscript „Full length transcriptome highlights the coordination of plastid transcript processing“ Guilcher and colleagues describe the successful usage of Nanopore sequencing to sequence complete plastid transcripts of Arabidopsis thaliana. Transcript versions differing in its maturation status are valuable to identify the order of maturation events (splicing and editing). Detected maturation events were compared with data of earlier studies to confirm the power of the method. With evaluation of long transcript sequences, the authors can not only detect different maturation intermediates of plastid transcripts, but also determine the dependence of different maturation events within one transcript.
The results presented in this study are interesting for the plant community as well as for the organellar biology community. The method is well introduced with all scripts and protocols available online. Authors need to ensure, that the webpages are also available in farer future, of course.
In my eyes, however, an enhanced presentation of data e. g. graphical representation would make it more convinient for the reader to follow the different aspects of the presented research project. I therefore suggest some aspects, which should be addressed prior publishing.
- In the introduction the authors describe the association between maturation factors in combination with the dependence of maturation events (line 40 and the following). Here I would suggest explaining the different reasons for dependence of maturation events in more detail: On one hand secondary structure effects of one maturation event can influence the other one (transcript based), on the other hand protein factors can influence different maturation events or can interact with each other (protein factor based). As additional aspect, one editing factor or splicing factor can also address several targets.
- Line 52/53: Please also cite Schallenberg-Rüdinger et al. 2013, Plant Journal in context of ccmFc editing of P. patens.
- Line 57: I would suggest writing „at the transcriptomic level“ rather than „at the genomic level“
- Line 75: The normal growth conditions should be defined, best here not only in the Material and Methods part.
- Line 78: The ligation step needs to be better introduced, what is ligated here.
- In the material and methods section, please also define the oligomer used for ligation.
- Abbreviations need to be explained (e. g. 1,55M, IGV)
- Information of split reads should be added (Fig. 1 or Material and Methods).
- Line 113: A figure of another publication Felder et al. is cited. Here I would suggest including important information of the former publication. In figure 2, probes used and data of the regular northern blot could be included.
- 3, colours of different sequence parts are not explained in the figure legend.
- Line 156: How did you calculate the R2 values in this context?
- Table 1, text and figures: To compare these new results with earlier results, it would be helpful to use a standardized abbreviation for introns and editing sites in the figures and tables of this paper as well as in general (intron nomenclature following Dombrovska et al. 2004 and editing site nomenclature following Rüdinger et al. 2009 for example). Editing sites should not be labeled with the genomic position, as sites are edited on transcript level. Labeling needs to be explained in each table and figure.
- Table 1: Why are 123 editing sites detected here? Are these listed somewhere and are these sites new editing sites or rather SNPs, which are often identified in NGS analysis. Did the authors also detect SNPs and exchanges except of C-U sites? Authors should comment on this.
- Line 168: Do the different intron splicing efficiencies reflect the biological impact of the genes? E. g. rps16-intron with only 4 % maturation rate?
- Figure 5 displays an interesting finding indeed. I would suggest using a line for the presentation of co-transcripts as well. Is it possible to add information of cleavage of multigenic transcripts?
- Line 250: An additional example: atp9 editing (atp9eU92SL) in patens mitochondria is dependent on splicing, as the target binding site is generated by exon-exon conjunction (Ichinose et al. 2013, already cited in the introduction)
- Line 286: The maturation of start codons as last step can also explain the often lower editing efficiency of start codon editing in comparison to editing events, which restore amino acid codons. The authors might wish to add that information (discussed in Small et al. 2020, Plant Journal)
- Line 290: It should be „…analysis of plastid transcriptome(s)“. It might be valuable to add „mitochondrial transcriptomes“ here as well, if the method can be used for both organelles in near future.

Reviewer 2 Report
Full length transcriptome highlights the coordination of plastid transcript processing
Guilcher et al. present a study applying a protocol for long-read sequencing of non-polyadenylated transcripts in Arabidopsis. The authors use the data generated by Nanopore sequencing to analyze RNA processing in the chloroplast transcriptome.
To my knowledge this is the first report of long-read sequencing of the chloroplast transcriptome. The authors analyze the data with respect to RNA editing and RNA splicing. Other RNA processing events, especially 3’ end processing, remain unanalyzed even though the library construction strategy is especially suitable for this analysis. The title promises a full-length transcriptome, which I think is not correct, since a relatively strong 3’ to 5’ bias and bias for short transcripts is evident from the presented data. In addition, the proposed interdependence of many processing events is in my opinion an overinterpretation of the co-occurrence of processing events and can be explained by differences in reaction rates for individual processing sites.
Major points:
- The authors claim there is only a weak 3’-5’ bias in the data and most reads were full-length. I don’t think this is correct and this has consequences for the analyses presented. The sequencing coverage for the gene ndhB drops to nearly 20% at the most 5’ editing site compared to the most 3’ site (Supplementary Table S1). The same trend is visible for rpoB and ndhD. This has consequences especially for the analysis of splicing, as shorter (spliced) transcripts should be overrepresented in the libraries compared to reads including two exons and intron. Accordingly, the splicing percentage is higher compared to the referenced Illumina dataset the authors compare their data with. Experimental evidence that the new protocol better represents the actual splicing status is missing. In this respect, I find it noteworthy that in figures 1 and 3 the authors do not show a coverage track and crop the alignment track. IGV by default orders reads according to the alignment start - thus this excerpt of the alignments are not a true representation of the data. The coverage tracks should be included in figure 1 and 3 and if the alignments are shown should not be cropped at the bottom.
Similarly, for Supplementary Figure S1 the authors use gene length rather than transcript length. The drop in coverage in the middle for long “genes” indicates they are overrepresented for genes containing introns. This analysis should be repeated using transcript length rather than gene length and a potential 3’-5’ bias should be re-evaluated and discussed. The title of the manuscript “full-length transcriptome…” should be changed accordingly.
- The authors interpret the statistical analysis of co-occurrence in a way that many processing events are interdependent. I wonder how two independent processing events with different reaction rates would be represented in this long-read sequencing. Wouldn’t the slower event look to be dependent on the faster reaction by default? In figure 5 the authors propose a chronology of maturation events for a number of transcripts. A relatively large number of transcripts for ndhD (25%) and ndhB (~50%) does not support the proposed chronology. I feel that alternative explanations for the observations are not sufficiently discussed in the manuscript. The title of the manuscript should be changed in this regard as well.
Minor points:
- The authors employ a protocol which uses amplification of the cDNA and amplification of the library by PCR. I’m surprised the authors did not use the information of the four random bases in the 3’ adapter to remove potential cDNA amplification PCR duplicates.
- A direct comparison of a Northern blot and the virtual Northern blot emulated from the Nanopore reads in Figure 2 would help to better understand the shortcomings of both techniques.
- Line 139-141: Felder et al. associate the 2600nt band with a transcript form including psbH. Felder et al. do not assign a band with unspliced petD, spliced petB.
- Lines 264-266: ndhD(117166) editing does not require editing at all other four sites in ndhD. In 71 out of 413 sequencing reads with partial editing ndhD(117166) is edited before the other four sites are completely edited.
- Lines 281-287: Too speculative… based on two transcripts the authors speculate that 5’ processing events are later than 3’ events. Except for the creation of a start codon I do not see how an unprocessed site would inhibit translation initiation.
- The library preparation should be included in the manuscript. The file deposited online is missing the information about the composition of a lysis buffer.
- How can the large variation between replicates in alignment rate with the chloroplast genome be explained?
